# Too Cute to Kill? The Need for Objective Measurements of Quality of Life

**DOI:** 10.3390/ani10061054

**Published:** 2020-06-18

**Authors:** Sarah Wolfensohn

**Affiliations:** School of Veterinary Medicine, University of Surrey, Guildford, Surrey GU2 7AL, UK; s.wolfensohn@surrey.ac.uk

**Keywords:** Welfare assessment, welfare legislation, quality of life, animal cuteness, public perception

## Abstract

**Simple Summary:**

Animal welfare is affected by how people think we should treat animals, and what people think is important affects the way they treat animals. The way people think about their animals is affected by how cute they perceive them to be. Animal welfare affects quality of life, which is about how sentient animals experience their lives, and understanding this requires animal welfare science. Ethics weighs that scientific knowledge with human desires (e.g., to earn a living by farming, to keep an animal as a companion, to clear land and build houses), so we decide how we should treat animals. The decisions that society has reached using science and ethical reasoning about how we ought to treat animals are reflected in animal welfare law. However, animal welfare law may often set a low standard of care, as there is little public money to enforce higher standards or the public is not aware of the welfare problems. The first step in ensuring a good quality of life is to measure welfare and then take action to improve it.

**Abstract:**

The recognition of animal suffering is influenced by cultural and societal prejudices and the cuteness of an animal leads to bias in the way it is treated. It is important to consider the animal’s behaviour and its environment—not just its physical condition—when assessing its quality of life. The Animal Welfare Assessment Grid (AWAG) is a useful tool for this purpose. The AWAG offers an evidence-based tool for continual welfare assessment, using technology where appropriate, such as digital activity recording, to facilitate decision-making and lead to improvements in the animals’ quality of life. It is highly adaptable to any species by assessing the four parameters of physical health, psychological wellbeing, environmental quality, and clinical and management procedural events. The outcome of assessing welfare should be action to improve it. Societal ethics and policy-making lead to legislation balancing the values we hold for different species. Influencing policy development in such matters as animal welfare, ecological conservation, and risks to humans requires a focus on public attitudes to, and understanding of, science, as well as consideration of potential unforeseen consequences of the social/environmental/economic impacts of policies.

## 1. Introduction

In veterinary companion animal practice, the health of the animal patient and its best interests are the main focus [1], and with the changing status of animals in our society, this patient-centred practice is becoming increasingly important as it has been recognised that there are increasing pressures in some circumstances that may override the patient’s interest in decision-making. Other possible motivations to carry out interventions can include professional advancement for the clinician, financial gain for the practice, a training opportunity for less experienced clinicians, and the potentially unrealistic expectations of the client (with no positive prospect for the animal) [2]. The human response is noticeably biased towards animals with particular features, which have then been actively selected for in the breeding of domestic pets. The baby schema system assesses a set of infantile traits that have a high appeal for humans. The term baby schema [3] refers to a set of facial features, including a large head and a round face, a high and protruding forehead, large eyes, and a small nose and mouth, which is found both in human and animal infants. This has been shown to affect cuteness perception and to elicit caretaking behaviour and attentional processes. Some breeds of both dogs and cats exhibit morphological and behavioural infantile characteristics which are retained into adulthood as a by-product of domestication [4,5] and selective breeding [4]. The presence of these lifelong youthful traits has been suggested as the basis of our attraction to pet animals [6] and cuteness judgment may enhance nurturing behaviour [7,8]. The response to an infantile facial configuration emerges early during development in children, and adult human positive appraisal toward animals informs therapeutic interventions involving pets [9]. The human selection for cuteness has compromised the physical health of some breeds of cat, dog and rabbit, and may also have made it more challenging to detect morphological indicators of pain and suffering, with the result that the cute pets may suffer more in various ways. The treatment of animals appears to be biased more by anthropocentric views than has been previously recognised, which may have a negative impact on their quality of life. There is therefore an increasing need for a tool to objectively assess quality of life and the potential or actual impact of planned interventions.

## 2. Recognition of Suffering

We may think that an animal is too cute to kill, but we must not leave them to suffer. The critical question about animals, as Jeremy Bentham (1748–1832) asked in 1789, is “not, Can they reason? nor, Can they talk? but, Can they suffer?” Then we must alleviate that suffering, and euthanasia may be the best option if we cannot improve quality of life. The decisions about when and how to end an animal’s life are perhaps the two most important ones in terms of managing its welfare, whether it is a companion pet, captive in a zoo, or wild and free ranging. For animals that appear cute this becomes more critical, as human emotion may delay us from intervening and preventing or ending the animal’s suffering. Any animal that is suffering severe pain, which cannot be alleviated, should be humanely killed immediately and not simply left to die.

This brings us to the dilemma of the recognition of suffering, which different animals manifest in very different ways. A critical issue now is to recognise what is welfare and to measure it in a quantitative manner to be able to demonstrate when welfare is deteriorating. This will ensure interventions can take place at an appropriate time to prevent suffering and to improve quality of life. Objective measures of quality of life are therefore required in order to make a decision about precisely when to carry out euthanasia, which is defined as the bringing about of a gentle and easy death (Oxford English Dictionary), an act which should not itself cause pain or suffering. Killing an animal is never a pleasant task, but it does not have to be unpleasant for the animal, provided it is carried out competently and humanely, although it may be an emotionally challenging experience for owners and care staff. The individual animal’s experience of pain and suffering is the same regardless of the context or ‘reason’ for its life, whether, for example, it is a pet animal, a farm animal, a wild animal, or an animal being used for research [10]. An animal cannot give informed consent and the potential benefits of continuing to live are easy to overestimate, whereas potential harms can be difficult to measure. The precise time at which to kill the animal must be based on appropriate and accurate clinical judgement. In order to demonstrate this, objective methods of assessing pain and distress in animals are necessary, and these are also valuable because they allow potential new treatment and intervention techniques to be evaluated critically. They also allow sound judgements to be made about the need for, and efficacy of, analgesics. Some veterinary surgeons and owners are often concerned that killing an animal too early might miss the opportunity to observe that a sick animal will indeed respond to and improve with a new treatment. It is crucial in such cases to identify signs that point to irreversible decline in the level of welfare.

Welfare has been defined according to the five freedoms (freedom from hunger and thirst; freedom from discomfort; freedom from pain, injury, and disease; freedom to express normal behaviour; and freedom from fear and distress) [11], the five needs (access to fresh water and a suitable diet that will keep them healthy; adequate shelter and somewhere comfortable to rest; access to veterinary treatment, but also steps taken to prevent pain, injury, or disease; company of other animals of their own kind with enough space and proper facilities so they can behave in a natural way, to be kept in conditions that mean they will not suffer; and to be treated in a way that does not frighten or distress them) [12], and the five domains (nutrition, environment, health, behaviour and mental state) [13]. These are essentially quite similar and reflect that welfare is more than simply the clinical condition of the animal and its health status. An animal can have good health and be productive and yet have poor welfare; for example, an intensively-kept laying hen may produce lots of eggs but it has a compromised quality of life due to the environment in which it lives, which does not permit a normal range of behaviours. Assessment of welfare needs to take into account the physical condition of the animal, its psychological condition and behaviour, the quality of the environment in which it is kept, and any procedures that take place on that animal even if they are being done for the animal’s benefit in the longer term. Since animals do not give informed consent and cannot understand the potential benefits of undergoing surgery, it is necessary to consider positive feelings (what an animal likes) and the resources it is motivated to obtain (what an animal wants) in order to determine the animal’s quality of life. The UK Farm Animal Welfare Committee in 2009 defined quality of life as a continuum from a life not worth living (with poor welfare and suffering) through an adequate life (a life worth living with minimal suffering) to a good life (with good welfare and mainly positive emotions and experiences) [14].

In order to reduce to a minimum the pain and distress experienced by animals, we need to consider: When could pain and distress occur? How can we detect or assess pain and distress? What can we do to reduce pain and distress? Unless there is evidence to the contrary, one can assume that interventions that would cause pain or distress in a human will also cause pain or distress in an animal. However, will it be the same amount of pain or distress? Will it last for the same amount of time? Will this be the same in all species? Before you can recognise what is abnormal, you must know what is normal, for the species, for the physiological state of the animal, and for that individual. It is necessary to contrast “normal” with what you normally see and what owners normally see. By making animals cuter by selective breeding and dressing them up, they may not actually be recognised as abnormal for the species and the impacts on their welfare may be missed (for example, the brachycephalic dog). Animals kept in deprived environments will be behaviourally and physiologically abnormal. Environmental enrichment is not an “optional luxury”, but enables an animal to be normal, behaviourally, physiologically, and immunologically.

Welfare concerns the quality of an animal’s life, rather than how long that life lasts (the quantity). Once an animal is dead, welfare is no longer a concern for that individual, but how an animal dies is a welfare concern and high mortality rates are often indicative of poor welfare. There may be an effect on dependent offspring and conspecifics, and social groupings may be disrupted which need to be taken into account and managed. When considering an animal’s quality of life, many people will also consider the value of that animal’s life [15]. However, should we consider the value to itself, or to science, or to food production, or as a pet/companion, or to its peer group? How is its value affected by economics, culture, politics, emotion, religion, or its cuteness? Does it actually have no value? Does it only have value because it is here for man’s use, or does it have inherent value simply by being sentient and having rational consciousness?

## 3. Cultural and Social Prejudices

René Descartes (1596–1650) said, “The greatest of all the prejudices we have retained from our infancy is that of believing that the beasts think. Animals have no capacity for reasoning, therefore no perception of pain”. Scientific evidence has now demonstrated that this is incorrect, and animals do have the capacity to reason and to perceive pain, thus answering Jeremy Bentham’s question of “Can they suffer?”. Children’s literature today encourages people even more to believe that animals are capable of rational thought. Immanuel Kant (1724–1804) said that “cruelty to animals is contrary to man’s duty to himself, because it deadens in him the feeling of sympathy for their sufferings, and thus a natural tendency that is very useful to morality in relation to other humans is weakened”. This opens up the now-proven fact that if we are nicer to animals, we will be nicer to each other. Should we therefore encourage the concept of cuteness in order to encourage being nice to animals, and potentially improving society?

Wider societal issues have impacted on animal welfare, starting with the Industrial Revolution and social upheaval as people moved into towns, away from the countryside, and animals became companions, rather than dinner or a workforce. The literature of the time reflected this, as shown in the popularity of Peter Rabbit [16] and Winnie-the-Pooh [17], and with this society began to show increasing concern for animal welfare. The cultural and social revolution and reduced exposure to human pain and suffering through medical advances led to an increased sensitivity to such matters. Publications such as Ruth Harrison’s *Animal Machines* (1964) [18], the Brambell Committee (1965) [19], and Jane Goodall’s studies of chimpanzees in Tanzania contributed to the rise of the anti-vivisection movement and the portrayal of “mad scientists” vs. cute and cuddly Peter Rabbit and friends. In the Victorian era, zoological collections were sourced from all around the world and the animals were simply put on display. Now zoos focus on conservation projects and education and are very conscious of animal welfare. Agriculture practices began to change in Europe and North America as farming became more industrialised, with a focus on production and efficiency to provide cheaper food for humans and to improve human health. Animals were housed in large numbers, which made for easier supervision, but increased disease and the need for vaccinations and treatment in order to maintain health and production. Animal welfare science began to develop to answer specific questions of public concern related to welfare, and organisations such as The International Society for Applied Ethology (founded 1966), the Universities Federation for Animal Welfare (founded 1926), and journals such as *Animal Welfare* and *Animals* were developed.

## 4. Ethics and the Law

Now in the 21st century, many people feel we have an obligation to animals, but for a variety of different reasons. Some consider that animals simply have intrinsic value; others feel this obligation because animals have value to us because we eat them and they are useful to us. For some, the obligation arises from the fact that animals can suffer or because the species is endangered. These considerations are reflected in ethics and the law, and there are different types of animal law reflecting differences in purpose such as protection of a species [20] or the environment, banning activities [21], prohibiting cruelty [22], controlling methods of production, improving animal welfare, improving public health, and improving animal health; which means they are covered by different Government departments and different animals are included. To determine the scope of a piece of legislation, it is necessary to first determine to which set of animals it applies: Are the animals specifically listed? Does it apply to domestic, wild, or captive animals, to vertebrate and invertebrate animals, or to all sentient animals? Though of course, whatever the use of the animal (pet, meat, laboratory, or wild), does the law matter to the animal? The animal’s perception of its welfare is not affected by the reason for its life or the cause of its suffering, whereas the human perception of its welfare is affected by the animal’s use and any intention to cause harm. There is muddled acceptance about the killing of certain types of animals and yet opposition to others. For example, society is very concerned about the use of animals in laboratories—which is aimed towards reducing suffering in people (and other animals)—but less concerned about the welfare of meat producing animals where the financial cost of the end product is the driving concern of some supermarkets, as reflected in their marketing strap lines (e.g., Tesco: “Every little helps”; Lidl: “Big on quality, Lidl on price”). There is also a cultural aspect contributing to these viewpoints—in the West, dogs are often perceived as cute, whereas in certain places in the East they are working animals or food. Horses, as another example, may be considered cute in the UK, but in France are considered food. Cuteness is a socio-cultural, and somewhat economic, creation.

Our obligation to animals as sentient beings, which is reflected in legislation and codes of practice and EU Directives, reflects society’s wishes for the overall good. We have a utilitarian approach to animal use, using the harm:benefit balance to justify whatever it is that we want to do to them—whether that is farming and eating them, riding them, watching them in zoos, or keeping them as companions and playthings. However, social attitudes are changing, for example, towards the environment and climate change and its impact on wild animals and agriculture. Animal welfare legislation has evolved over centuries, influenced by cultural views. However, does legislation work? Sometimes it may not be effective in ensuring animal welfare because of the legal status of animals as property, their sentience may not be recognised, strict liability offence or intent is needed, there are religious and cultural exemptions, and some things are just classified as “accepted practice”. Sometimes there are compromises in wording, in which the original meaning and intention is altered or reinterpreted by industry bodies, or there is subjective interpretation by courts or juries, or the text is sometimes simply poorly worded. There may also be a lack of enforcement, because responsibility for enforcement is spread between different government departments with limited resources. There may be limited powers and conflicting priorities of the enforcement bodies. For example, the inspection of an animal, when there is suspected cruelty or lack of welfare provision, requires access to premises for inspection, but privacy laws must also be respected. Animal experimentation is required to test medicinal products for safety, and this too has to be balanced against the welfare of the animals. Added to which, there may be commercial confidentiality and the decision must be made as to whether prosecution is in the public interest. All this is before evaluating the animal’s cuteness.

The ideal welfare law applies to all sentient animals, is clearly written, includes failing to meet an animal’s needs among the offences, is easy to amend in line with new scientific knowledge and ethics, has high legal status which allows for prosecutions, has a clear enforcement responsibility, involves an enforcement body with sufficient power and funds, and includes education of the public and industry. Animal welfare is a complex concept. Understanding it requires science (how different environments affect an animal’s health and feelings, from the animal’s point of view) and deciding how to apply those scientific findings involves ethics (how humans should treat animals). Enforcing those decisions in society involves the law (how humans must treat animals). There needs to be greater emphasis on ways of improving animal welfare, such as financial incentives for producers, public education, including animal welfare in the school curriculum, and research to inform government policy. Maintaining a high standard of animal welfare is essential wherever animals are managed.

## 5. Measurement of Welfare

Animal welfare needs to be measured, and if welfare is identified as poor, the cause should be found and a solution should be identified. Welfare assessment is much more than the presence (or absence) of health or disease. It allows assessment of responses to treatments or environmental changes that affect animal welfare. The measurement of animal welfare must lead to action aimed at improving it. Welfare can be assessed using a combination of resource-based and animal-based indices, often divided into behavioural indicators, physiological indicators, and clinical/pathological signs. To assess the animal’s welfare, it is necessary to have in place objective measures of suffering or of well-being, such as those described in Main et al. [23]. Modern animal welfare assessments should aim to evaluate positive as well as negative states. Within the literature there is huge variability about the indicators that are useful in assessing these states for some species. Optimising welfare assessment using life in the wild as a baseline is not necessarily the most accurate indicator of what is right for an animal in captivity, which makes the identification of factors to include within species-specific welfare assessment even more challenging. There is no “one size fits all” welfare strategy, as it should account for the range of biological requirements and needs of the species. It should be carried out in an objective and consistent manner, not in an ad hoc impression-based way, involving the whole animal care team at key life history points, such as weaning, relocation, during training, and at clinical intervention. Assessment of welfare is not simply about recognising pain. The harms that the animal experiences relate to all five domains of animal welfare, i.e., nutrition, environment, health, behaviour, and mental state. A key element of ensuring good welfare will be to provide an environment that allows the animals to express their natural repertoire of behaviours as far as possible [24]. Welfare assessment should therefore include consideration of the environment and the behaviour of the animals, not simply their physical condition. The data may involve the use of scoring sheets and checklists, and should account for species differences, age or developmental differences, differences in sex and reproductive status, and differences in social status.

An animal’s experiences of its environment and resources, such as feeding, handling, and transportation, create sensory inputs that its brain evaluates, resulting in emotions such as fear or pleasure. Its body responds to those feelings and emotions with physiological and behavioural responses, which enable the animal to adapt and survive, but which impact on its welfare status. Ethology, the scientific study of animal behaviour, has developed methods of examining in detail the different categories of animal behaviour; for example, for reproductive behaviour, courtship, mating, and maternal behaviour; and for feeding behaviour, seeking, obtaining, storing, and consuming food. Within each category, each species has typical behavioural patterns, such as predators using their senses to detect prey, then the chasing, killing and the consumption of the dead prey may follow a particular pattern within the social group. How animals allocate their time to different behaviours (the use of time budgets) is examined by making detailed behavioural observations over many days to establish the relative amounts of time that animals allocate to performing different behaviours within the categories. This can then be used to quantify how animals allocate their time in specific conditions. Some specific behavioural indicators of poor welfare include a limit in the range of activity, such as no play or no response to external stimuli; sickness, pain, and depressive behaviours; abnormal fear or aggression towards humans; fighting and breakdown in social hierarchies; and stereotypic behaviours.

Behavioural indicators of good welfare will include various behaviours and a range of activities indicating that the animal is alert and curious about its external environment and social hierarchy; for example, exploration, play, interaction with other members of the group/herd/flock, interaction or avoidance of humans, aggressive behaviours, and affiliative behaviours such as allogrooming. However, behaviours are affected by many factors. These include whether the animal is a prey or predator species (since prey species have evolved to hide pain and infirmity); the breed of the animal (e.g., pit bull terrier vs. labrador); its age (young animals are more active, more likely to play, but spend more time sleeping); sex; social/reproductive status; group size and interaction (the presence of dominant male and young males); dominance hierarchy (maternal rank in some species); and season (breeding, migration). It is critically important to understand the animal’s behavioural repertoire—what is normal for the species, breed, sex, etc., and what is normal behaviour for an individual—remembering that normal is not necessarily what you normally see. The behaviour will be affected by the available environment and limited to what is available within a pen. Some species require a more complex three-dimensional environment, as they need to be able to climb. Dogs may prefer to run free but may always be kept on a lead. Wild free ranging animals may have adapted to new urban environments.

Welfare measurement can incorporate an analysis of production records which are available from routine management information, such as longevity, growth rate, susceptibility to disease, productivity, and infant care. Others require individual assessment, such as wound healing, coat and body condition, body shape, and posture. Information requiring assessment of behavioural repertoire and activity budgets, such as grooming, sleeping, play, social behaviours, facial expressions and vocalisations, can be obtained by observation of video footage of animals undisturbed by a human presence in their environment. The affective component of pain in animals can be assessed by measuring the pain face using facial expressions. Such scoring systems have been described for various species [25,26,27]. Assessment of the physiological/biochemical conditions can be carried out by measuring hypothalamic–pituitary–adrenal axis hormones in blood, faeces, saliva, tears, or hair, or immunological functions such as lymphocyte activity [28]. Heart rate, blood pressure, and body temperature measurements can also be used, but some of these parameters will require instrumentation and invasive techniques, which may be detrimental to the animals’ welfare.

## 6. Quality of Life Assessment

Welfare can be defined by application of the five domains of nutrition, environment, health, behaviour, and mental state, but is usually only assessed at single moments in time and does not take into account the duration of any adverse welfare impact. Most schemes only record the detailed welfare state of the animal within a relatively short time frame, at one particular moment, or over the duration of a particular treatment, which is generally very short in proportion to its lifetime. These schemes rarely reflect cumulative suffering and the lifetime experience of the animal, which should be evaluated in addition to the welfare at individual moments in time. Combining a range of assessment parameters into one usable tool has been identified as an important goal in providing practical, objective, and robust assessments of lifetime experience and welfare in order to be able to use this to determine when it is appropriate to end an animal’s life. The Animal Welfare Assessment Grid (AWAG) offers an evidence-based tool for continual welfare assessment, using technology where appropriate, such as digital activity recording, to facilitate decision-making and lead to improvements in the animals’ quality of life. The AWAG tool encompasses the five domains of animal welfare and draws attention to the temporal component of any suffering that is often overlooked [29], reflecting the cumulative lifetime experience of an individual. This an intrinsic benefit of the tool, that does not exist in other welfare measures. It is highly adaptable to any species, scoring four parameters that impact on well-being, namely, physical health, psychological wellbeing, environmental quality, and clinical and management procedural events. Within each parameter, a number of factors are evaluated to give a numerical score, which is then used to give a total welfare score for that particular time-point. The AWAG has been tested using both retrospective and in-life data and has been found to give a clear indication of animals’ welfare during their lifetime. Within each parameter (physical, psychological, procedural, and environment) of the Grid, various factors are scored between one and ten. Each factor score is defined using descriptors for each number to reduce scoring bias. For example, the physical score would encompass the animal’s general condition, clinical assessment, pain control, inappetence, activity level, etc. A score of one indicates the best possible state (lowest possible impact on welfare), whereas a score of 10 would be the worst possible state (highest possible impact on welfare). For each parameter, mean factor scores are calculated by the software and used to create a polygon, the total area of which is calculated to derive a cumulative welfare assessment score at that particular point in time [30]. In addition to the ability to quantify quality of life at a given time-point, the tool’s software provides a visual representation of the animal’s welfare state and can be used to drill down to show which factors have contributed to that state (see Figure 1 and Figure 2). If these are identified as sub-optimal, specific intervention can then be undertaken to improve the animal’s well-being, targeted at specific elements.

Thus, the software provides a visual and numerical summary for individual animals or groups of animals, which can be interrogated to provide evidence of improving or deteriorating welfare, empowering people to take action from a position of knowledge and with an evidence base to improve welfare. The objective nature of the tool will ensure there is no owner bias or distraction by the cuteness of the animal or its potential economic value, assisting decision-making about possible interventions. The outcome of using the AWAG will therefore allow objective feedback on changes affecting animal welfare and objective evaluation of the animal’s quality of life and any cumulative suffering. Continued monitoring will then demonstrate improvements in welfare, since the outcome of measurement should be action to improve welfare.

Extensive studies have been conducted using the AWAG in a variety of settings demonstrating that it is very transferrable and can be used to manage the welfare of a wide range of species. It has been successfully applied in settings where evaluating and improving animal welfare is important; for example, zoos, veterinary clinics, wild-life trusts, and rescue centres [31,32]. By combining retrospective and prospective analyses of welfare, the AWAG can assist in making difficult quality of life decisions about an animal or group of animals. Assessment of animal welfare is essential, not only to meet legislative requirements but also to identify areas where the experience of an animal can be improved in terms of environment. The AWAG can provide powerful documented evidence of why changes are required or how improvements have been made. This evidence can be used to justify changes to techniques and expenditure on resources and is invaluable in presenting positive changes to housing; management; or interventions to regulators, review bodies, and to staff and visitors. The AWAG software is an open source project available to the wider community via https://github.com/PublicHealthEngland/animal-welfare-assessment-grid/wiki. It is designed to run on a server with users accessing it via a web browser on their PC, laptop, or mobile device. As such, it currently requires technical IT knowledge to install and configure it. The initial funded project was completed in January 2020 and further development of the tool is now underway to improve the ease of installation and setup of the software, making it more accessible for individual users and small companies. The user interface will be updated to ensure maximum usability and an intuitive, user-friendly layout with streamlining of the data entry and interoperability of the system, to enable interaction with existing animal information management systems and avoid duplicating the data input on other systems.

There are other commercially available systems for handling general data related to animal management, but these systems do not extend to capturing welfare data in the way that the AWAG does. These systems are designed to produce reports on animal usage, training, and staff competency, for example. Incorporating the AWAG tool into these existing systems would add the ability to evaluate and quantify animal welfare in such a way that the data generated can be easily used, and it will be developed to use technology where appropriate, such as digital activity recording from precision livestock farming techniques. Animals need to be monitored with sufficient frequency, guided by thorough knowledge of the normal behaviour and physiology of the species, and the use of appropriate monitoring and assessment tools, as described earlier. Developments in non-invasive technologies, such as imaging, 24-h remote monitoring of behaviour, and movement detection by automated recording are becoming more sophisticated. Use of these technologies and physiological monitoring via biotelemetry may enable pain and distress to be avoided by identifying non-clinical criteria that occur prior to observable suffering or clinical manifestations of a condition, thus providing opportunities to improve welfare and quality of life.

## 7. The Outcome of Assessing Welfare

The outcome of assessing welfare is that it will allow feedback on changes affecting animal welfare, it will encourage communication about animal welfare between care staff, owners, regulators, funders, retailers, and the public, while accounting for the justification of any required methodology of treatments/housing/care, etc., which impact on welfare. The evaluation of the animal’s quality of life should be carried out prospectively to assess the potential effects of a planned intervention on an ongoing basis and to assess what is really taking place. It should also be carried out retrospectively in order to be able to review the events and change the management strategy for the next time. The challenge for animal carers is to accurately interpret the state of the animal from the information available and not to be swayed by the level of cuteness. If any doubt exists about how to interpret its behaviour and clinical signs, then the maintenance of good welfare of the animal must come first, and the responsibility for this lies with the owner (under the Animal Welfare Act) and with the attending veterinarian (as defined in the Royal College of Veterinary Surgeons’ declaration that “my constant endeavour will be to ensure the health and welfare of animals committed to my care” [1]). It is important to remember that the assessment of welfare alone is not sufficient. If welfare assessment is simply an academic exercise, it does nothing for the animal. Proper welfare assessment allows feedback on changes affecting the animal’s welfare and then decisions must be made about what action should be taken. The outcome could be to do nothing; but if so, there must be clearly-articulated reasons about why nothing is done. The entire team must accept responsibility in delivering animal welfare; monitoring is simply the tool that demonstrates whether this delivery is being achieved. Decisions on euthanasia must be reached swiftly and appropriate actions taken promptly if suffering is to be prevented, no matter how cute the animal may appear.

Advances in veterinary medicine mean prolongation of life is more easily achievable than in the past and have coincided with the companion animal’s transition to family member status, encouraged by the development of cuteness. In modern veterinary medicine, there are numerous treatment options [2], many of which lack evidence-based practices [33]. As a result of this, there is increasing concern that these options are encouraging overtreatment, and whether these modern and innovative potential treatments consider the best interests of the animal [34]. Veterinary surgeons encounter many difficult decisions and regularly face ethical dilemmas [35,36]. Stress ratings are highest when there is conflict between the welfare of the animal and the interests of the client [37]. The vet’s role has now moved beyond the short-term medical care of the animal to include long-term management of the emotional well-being of the attached human carer [38], who may prolong the animal’s suffering by refusing to consider euthanasia as they come to terms with guilt and anticipatory grief. This delay as the vet waits for the client to psychologically accept the pending loss of the animal puts the vet into a pivotal, if not conflicted, position [39], and leads to moral stress in veterinary practice with calls for improved decision-making tools [40]. Currently, there are no tools available to assist veterinarians in the decision-making process that prioritise the overall quality of life, rather than just the quantity of life. Veterinary surgeons, unlike doctors, are very privileged to have euthanasia as a treatment option for their patients in order to reduce potential suffering, but it is often perceived as indication of failure to make the animal better. Every animal will die at some point, and expert management of a timely and stress-free death is part of the veterinary surgeon’s role. How that is perceived by owners can have a significant impact on the vet. Further development of the AWAG and its software will assist vets in their decision-making related to chronic disease, surgery, and prolonged medical treatment by quantifying the risk and benefits of treatments for an individual animal, thus preventing therapy that compromises welfare. It will enable clinicians to make well-reasoned, ethically justified decisions, reducing the influence of the animal’s cuteness, thus promoting positive clinical experience and assurance for owners, which is of critical importance in veterinary practice. By enhancing the quality of decision-making, decisional conflict will be reduced, thereby improving mental health in veterinary surgeons. The use of the AWAG can assist with client management, because it provides a clear visual illustration of quality of life and thus supports the vet’s decision making and helps owners to understand the reasons for the vet’s advice.

If welfare is identified as sub-optimal, then the cause should be found, a solution should be identified and implemented, and then the welfare should be reassessed to ensure it has improved. The important message being: take action. Otherwise, why measure it at all? Monitoring does nothing for the animal. Is the measuring of welfare simply an academic exercise? The outcome of measurement must be action to improve welfare. Sometimes the decision will be to do nothing, but it is necessary to justify why this decision has been made. Understanding animal welfare and how different environments affect the animal requires science. Deciding how to apply those scientific findings involves ethics. How should humans treat animals? Who decides, and how do we balance different opinions? Whatever our ethical position, surely this should be applied to all use (or abuse) of animals. Some people consider it immoral to eat meat, to wear leather or furs, to hunt, shoot or fish, to go to the circus or to the zoo, or to use animals for research, but many people have ambivalent attitudes. They consider it immoral to eat red meat but not white meat, even though intensively-produced chicken may well have worse welfare than extensively-produced lamb or beef. They may poison or trap rodents as vermin but keep others as pets. Many people keep cats as pets, but they kill >100 million wild animals/year in the UK [41]. Do these opinions depend on the cuteness of the different species?

## 8. Conclusion: Balancing Animal Welfare and Human Perceptions

Animals’ inherent value cannot be compared to the inherent value of humans as moral agents, because animals cannot respect humans’ rights. Most people accept that we can kill animals for our own ends such as food, clothes, etc., although the animal rights theory forbids this. However, while the majority of people do not grant animals the right not to be killed, they do accept that animals might have the right to be protected from suffering, and much European legislation about animal welfare offers this protection to ensure good animal welfare. The legislation follows the utilitarian philosophy, distinguishing between a right and wrong action by its consequences in relation to happiness or pain, and the ‘rightness’ of an action is determined by its contribution to the happiness of everyone (or everything) affected by it: “The greatest good for the greatest number”. Therefore, in any situation one should choose the possibility which most augments the general welfare. However, this raises yet more questions: How do we make that calculation? How do you compare human happiness and animal happiness in the same equation? How is human happiness affected by animal cuteness? And what evidence can you use when bringing breaches of the legislation to court? The AWAG has been cited as a potentially useful tool to assess the welfare of animals and to support organisations such as the RSPCA to intervene earlier, to work in partnership with owners and to support claims of risks to animal welfare when investigating and prosecuting animal welfare offences [42].

Finally, how do these considerations affect the complexity of policy development and implementation by governments, and on environmental, public, and veterinary health and welfare? Policy makers need access to ethical and sound evidence on which to base the policies, which must be robust enough to benefit social well-being and societal progress. People need to be part of the decision, but will not necessarily accept ‘expert’ opinion. Indeed, sometimes experts ask the wrong question, or don’t define it clearly and thus get opposing answers which then undermine the value of their expertise. Sometimes all the evidence and information that is needed is simply not available as it may take time to collect, but decisions by policy makers may have to be taken urgently. Social media has a significant influence; the level of cuteness of animal stories reported has an impact, and the financial model of newspapers has changed so they are less able to investigate and subject issues to ethical review, as this will slow down the process of getting into print. Social scientists doing research are subject to institutional ethical review bodies, but journalists are not—why the difference? Should the free press be more accountable, since the media plays a significant role in confusing issues around animal welfare, with a tendency to highlight the owner’s view or the animal’s cuteness? A single solution to promote higher animal welfare across the entire EU is unlikely, due to significant regional differences and the fact that what is appropriate in each region depends on many factors [43]. Influencing policy development in such matters as animal welfare and biodiversity requires animal sciences (with a focus on public attitudes to, and understanding of, science), and also depends on the relevance of these issues with policymaking and their resonance with public perception. Influenced by the concept of cuteness, the values we hold for different species translate into policy-making, via ideas around animal welfare, ecological conservation, and risks to humans. However, there may be unforeseen consequences of the social/environmental/economic impacts of policies, and one needs to be mindful of the interaction between the systems of the natural world and humans.

## Figures and Tables

**Figure 1 animals-10-01054-f001:**
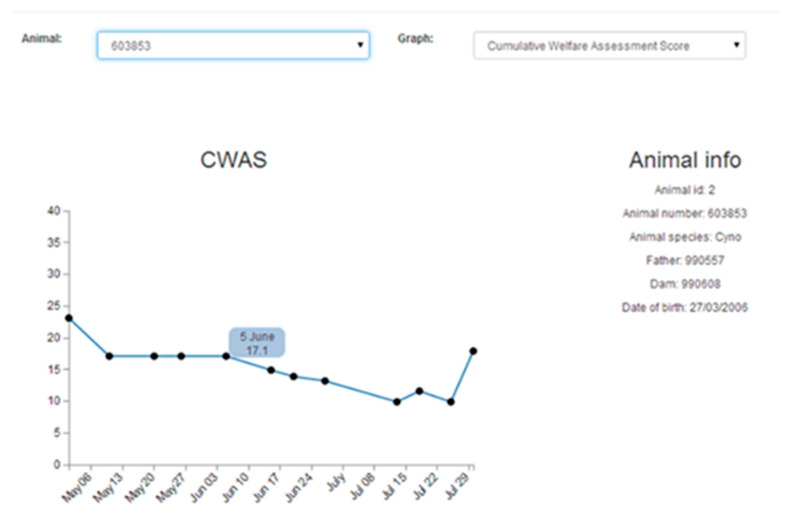
The Animal Welfare Assessment Grid (AWAG): Cumulative welfare assessment score graph.

**Figure 2 animals-10-01054-f002:**
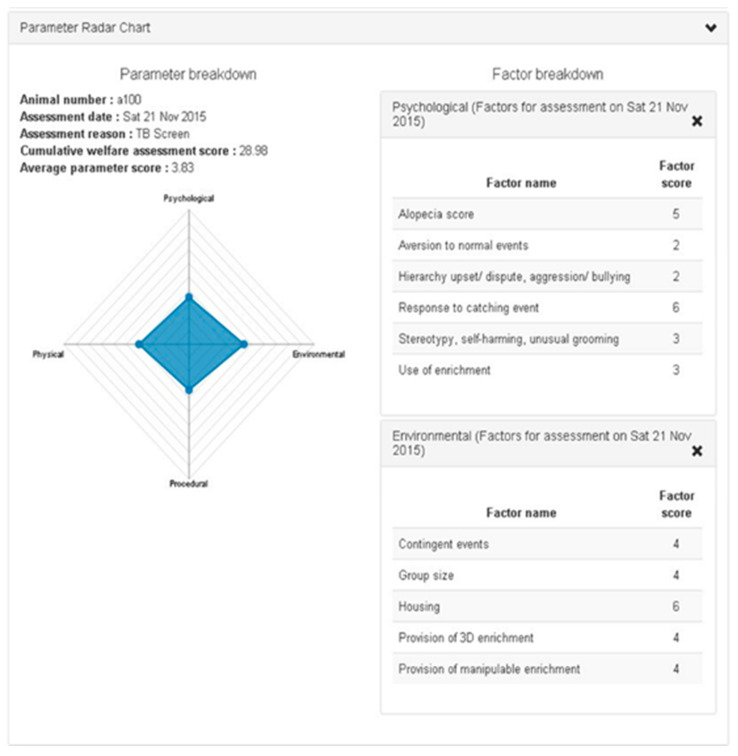
The AWAG: Clicking on the cumulative welfare assessment score (CWAS) at one point brings up the Grid for that moment in time and the detail of the factors contributing to that welfare assessment.

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
