# Peer review of "Too Cute to Kill? The Need for Objective Measurements of Quality of Life"

_animals, 2020, doi:10.3390/ani10061054_

Round 1
Reviewer 1 Report
Thank you very much for allowing me to review the article "Too cute to kill, but quality of life is what matters" (animals-831400).
It is an interesting commentary, worth of publishing. However, fter reading the manuscript, one can say that one of the main objectives of this commentary was to familiarize the readers with the AWAG grid. Some information about this tool should be added to the abstract. In manuscript, definitely some additional information about this tool/software are required – it is a freeware/shareware tool, non-profit ?; homepage, name of institution/developers.
Just a few minor comments.
L52 Please name these five freedoms and five needs.
L56 While this is generally correct, please consider changing “intensively kept broiler chicken” to “intensively kept laying hens”
L106 Not all international readers are well-familiarized with British children literature. Please give the year of publication.
L120-121 Please give the year of foundation of organizations.
L144-145 East, West (with capital letter) ?
While the work is very well written, some editorial corrections are required throughout the whole manuscript.
Author Response
Notes for reviewer 1
Thank you for your positive comments
Some information about this tool should be added to the abstract. see lines 22-25.
In manuscript, definitely some additional information about this tool/software are required – it is a freeware/shareware tool, non-profit ?; homepage, name of institution/developers. See lines 351-360
L52 Please name these five freedoms and five needs. Line 88-95
L56 While this is generally correct, please consider changing “intensively kept broiler chicken” to “intensively kept laying hens” line 98
L106 Not all international readers are well-familiarized with British children literature. Please give the year of publication. line 148
L120-121 Please give the year of foundation of organizations. line 162-163
L144-145 East, West (with capital letter) ? line 186-197
Reviewer 2 Report
The premise of this commentary article centres around the idea that how people think of animals influences the way they perceive them and their needs, and ultimately how they treat them. Much of the article is occupied with discussion around recognition of suffering and how to assess animal welfare. The author draws on animal welfare science, welfare law and ethical frameworks to support this discussion and describes a welfare assessment tool developed by the author and colleagues. The need for animal welfare assessment to result in an active response to improve quality of life is emphasised.
How animals are perceived and the implications this has for decisions surrounding their care and quality of life is an intriguing and complex issue and the idea behind this article is good. However, the title of this commentary, ‘Too cute to kill’, promises a more detailed consideration of this topic than this article provides. Essentially the article provides a review of animal welfare assessment. The concept of cute is not defined or explored. The article would greatly benefit from an opening paragraph outlining what cute it, how perception of cute may vary with individuals, that some companion animals in particular have been selectively breed for cuteness and the impact this has had on their welfare. All of these issues need to be introduced to put the rest of the article in context. These concepts then need to be expanded throughout the remainder of the article and the topics raised need to be viewed through the lens of how what you are saying relates to the idea of being too cute as a potential risk for poor quality of life. At present, this article is not coherent and only fleeting consideration is given to the consequences of being cute outside of the article title. Consequently, the article does not currently meet the requirements for a Commentary paper to be an in-depth, novel way of interpreting the literature.
Author Response
Notes for reviewer 2
I have added an opening paragraph outlining the features that affect perception of cuteness, that some companion animals have been selectively bred for cuteness, the possible impact this has had on their welfare, and cited relevant references, see lines 35-56. The concept of cuteness and social perceptions of animals are referred to throughout the document (specifically at lines 66, 132, 145, 156, 189, 212, 278, 338, 387, 399, 402, 443, 469, 474)
Round 2
Reviewer 2 Report
I am in agreement with Reviewer 1, that the main objective of this paper seems to be to familiarise readers with the AWAG tool, rather than a true reflection on the impact of cuteness on animal welfare and quality of life. The addition of a new paragraph in the introduction has benefited the paper in this respect, bringing cuteness as referred to in the title more to the forefront of the paper. However, I honestly think that the title of the paper does not reflect the content, even with the additional paragraph and the mentions of cuteness in the text. I suggest that the title is changed to something more representative of the content e.g. "Too cute to kill? The need for objective measurements of quality of life" or similar.
Minor comments:
Opening paragraph valuable addition, thank you. You could consider mentioning how human selection for cuteness has compromised physical health in some breeds of cat, dog and rabbit, and may also have made it harder to detect morphological indicators of pain and suffering. Our cute pets may suffer more as a result in numerous ways. Focusing on baby schema here makes the following paragraph, when you mention captive wild and free-ranging animals, seem a bit incongruous so it might be worth broadening this discussion to include human responses to animals with particular features of all species (baby features, mammals tend to be preferred over cold-blooded animals and birds etc), which we have then gone on to actively select for in our domestic pets, at which point you introduce baby schema.
Line 162-163: May be better to rephrase sentence for clarity e.g., Animal welfare science began to develop to answer specific questions of public concern related to welfare and organisations such as The International Society for Applied Ethology (founded 1966), Universities Federation for Animal Welfare (founded 1926), and journals such as Animal Welfare, Animals were developed.
Line 416-425: Bring in the value of having an objective tool that won’t be influenced by the animals’ physical appearance/cuteness
Line 441-480: The concluding comments needs greater discussion around cuteness than it currently has - a one-off mention in Line 471 in relation to the media
Author Response
Thank you for your comments and the prompt response.
I have changed the title as suggested.
See lines 55-58 for a mention of how human selection for cuteness has compromised physical health in some breeds of cat, dog and rabbit and lines 43-45 on the human response towards animals with particular features and the breeding of our domestic pets.
Line 162-163: Now 165-168 – I have rephrased as suggested.
Line 416-425: Now lines 425-426: I have included a comment on the animal’s cuteness to enhance the value of the AWAG tool
Line 441-480: Now lines 458-483: See additions in lines 458-459 and line 483.